# Genetic Characteristics of *Acinetobacter baumannii* Isolates Circulating in an Intensive Care Unit of an Infectious Diseases Hospital During the COVID-19 Pandemic

**DOI:** 10.3390/pathogens14100961

**Published:** 2025-09-23

**Authors:** Svetlana S. Smirnova, Dmitry D. Avdyunin, Marina V. Holmanskikh, Yulia S. Stagilskaya, Nikolai N. Zhuikov, Tarek M. Itani

**Affiliations:** 1Federal Scientific Research Institute of Viral Infections «Virome» Rospotrebnadzor, Letnyaya Street, 23, 620030 Yekaterinburg, Russia; avdyunin_dd@niivirom.ru (D.D.A.); stagilskaya_ys@niivirom.ru (Y.S.S.); zhuikov_nn@niivirom.ru (N.N.Z.); itani_tm@niivirom.ru (T.M.I.); 2The State Autonomous Healthcare Institution of the Sverdlovsk Region “City Infectious Diseases Hospital”, Sulfatnaya Street, 4, 622005 Nizhny Tagil, Russia; gib-nt@mail.ru

**Keywords:** *Acinetobacter baumannii*, whole-genome sequencing, antimicrobial resistance, MLST, cgMLST, COVID-19, ICU, hospital environment

## Abstract

During the COVID-19 pandemic, a significant increase in the spread of healthcare-associated infections (HAIs) and antimicrobial resistance (AMR) was observed. *Acinetobacter baumannii*, particularly carbapenem-resistant strains, poses a serious threat in intensive care units (ICUs). This study aimed to genetically characterize *A. baumannii* isolates from the ICU of an infectious diseases hospital repurposed for COVID-19 patient treatment. Whole-genome sequencing (WGS) was performed on 56 *A. baumannii* isolates from patients and environmental surfaces using the Illumina MiSeq platform. Bioinformatic analysis included multi-locus sequence typing (MLST), core-genome MLST (cgMLST), phylogenetic analysis, and in silico detection of antimicrobial resistance genes. Three sequence types (STs) were identified: ST2 (35.7%), ST78 (30.4%), and ST19 (3.5%); while 30.4% of the isolates were non-typeable. Phylogenetic analysis revealed clustering of ST2 with isolates from East Africa, ST78 with European isolates, and ST19 with isolates from Germany and Spain. Resistance genes to eight classes of antimicrobials were detected. All isolates were resistant to aminoglycosides and β-lactams. The *bla*OXA-23 carbapenemase gene was present in all ST2 isolates. cgMLST analysis (cgST-1746) showed significant heterogeneity among ST2 isolates (24–583 allele differences), indicating microevolution within the hospital. A novel synonymous SNP (T2220G) in the *rpoB* gene was identified. Environmental sampling highlighted the role of contaminated personal protective equipment (PPE) in transmission, with 47.0% of ST2 and 64.3% of ST78 isolates found on PPE. The study underscores the high resolution of WGS and cgMLST for epidemiological surveillance and confirms the critical role of infection control measures in preventing the spread of multidrug-resistant *A. baumannii*.

## 1. Introduction

The increasing threat of epidemic-prone infectious diseases is evident from recurrent global outbreaks of emerging viral infections, the rising incidence of healthcare-associated infections (HAIs), and the spread of antimicrobial-resistant (AMR) microorganisms [1]. The prevention of HAIs remains a pressing issue, requiring new approaches in the context of emerging biological threats. In the Russian Federation, the COVID-19 pandemic necessitated the rapid deployment of infectious disease hospitals with segregated “red” and “green” zones, which often led to disruptions in established HAI prevention protocols [2].

These hospitals created dynamic, closed ecosystems where patients underwent numerous invasive procedures and received antimicrobial treatment, fostering the active circulation of pathogens, their adaptation, and the selection of resistant strains [3]. The spread of HAIs and AMR is a recognized contemporary biological threat, challenging patient safety, healthcare workers, and public health [4,5]. Environmental microorganisms can acquire resistance mechanisms and become clinically significant opportunistic pathogens (OP) upon entering hospital ecosystems [6,7].

*Acinetobacter baumannii* is a prime example—a ubiquitous environmental bacterium that has evolved into a significant OP in healthcare settings [8,9]. Carbapenem-resistant *A. baumannii* (CRAB) is a leading cause of nosocomial infections in ICUs [10,11]. The COVID-19 pandemic was associated with numerous hospital outbreaks of *A. baumannii*, posing a severe threat to patients [12,13].

Transmission routes in healthcare facilities are diverse, including direct contact, contaminated surfaces, medical devices, and notably, aerosols and personal protective equipment (PPE) [14,15]. *A. baumannii* sequence type 2 (ST2) is a successful, globally distributed clonal lineage associated with outbreaks and multidrug resistance (MDR). Classical MLST, based on seven housekeeping genes, lacks the resolution to track the microevolution of *A. baumannii* within a hospital [16].

High-throughput sequencing methods are essential for accurate epidemiological diagnostics, enabling high-resolution typing, relatedness assessment, and detection of resistance and virulence determinants [17,18,19]. This study aimed to provide a genetic characterization of *A. baumannii* isolates from an infectious disease hospital ICU during the COVID-19 pandemic using whole-genome sequencing data.

## 2. Materials and Methods

### 2.1. Sampling and Bacterial Isolation

From 2022 to 2023, biological and environmental samples were collected from the Nizhny Tagil’s City Infectious Diseases Hospital (Nizhny Tagil, Russian Federation) for treating COVID-19 patients for the presence of OP. These samples were collected in accordance with a scheme for surface swabs patented by the authors, which was developed for simultaneous assessment of both viral and bacterial contamination [20]. The samples were collected using two sterile cotton swabs moistened with 0.1% peptone water containing neutralizers of disinfecting agents for three days every four hours at 20 sampling points from environmental surfaces grouped into three blocks: PPE of medical workers (the outer surface of the doctor’s/nurse’s/orderly’s PPE suit, the outer surface of the upper pair of medical gloves: doctor/nurse/orderly), patient care environment (the surface of medical manipulation table, the outer surface of the medical syringe dispenser, handrails and adjustment levers on an ICU bed, the outer surface of the ventilator), general hospital sampling points (The outer surface of the suction unit, dispensers for liquid soap and hand sanitizer, ICU door handles, oxygen pipeline valves, ICU electric light switches, clinical doctor’s workspace) (Figure 1). A detailed list of each isolate, including its sample ID and the specific sampling point it was obtained from, is provided in Appendix B Table A1.

### 2.2. Bacterial Isolation and Cultivation

To isolate *Acinetobacter* from clinical specimens, sputum samples were used. For environmental surveillance, surface swabs were collected from the hospital environment.

Surface swabs were collected from various objects using sterile cotton swabs moistened with 0.1% peptone water containing neutralizers of disinfecting agents.

Sample collection and processing were performed in accordance with the national methodological guidelines MUK 4.2.2942-11 “Methods for sanitary and bacteriological testing of environmental objects, air, and sterility control in healthcare institutions”, which are effective in the Russian Federation.

To isolate the genus *Acinetobacter*, the swab fluid was inoculated into tubes containing synthetic ethanol-ammonium medium and incubated at 30 °C for 48 h. After enrichment, samples were streaked onto selective agar and incubated at 30 °C for 24–48 h.

Colonies presumptively identified as *Acinetobacter* were selected for further analysis. Gram staining was performed to identify Gram-negative rods or coccobacilli, and an oxidase test was conducted. Oxidase-negative, Gram-negative cultures were subcultured onto meat-peptone agar (MPA) to obtain pure cultures for transport.

### 2.3. DNA Extraction and Whole-Genome Sequencing

DNA extraction was performed using the «RIBO-prep» reagent kit (FBIS CRIE, Moscow, Russia) from 24-h cultures grown on MPA following the manufacturer’s protocol. DNA concentration was assessed using a Qubit 4.0 fluorometer and the HS Qubit™ dsDNA HS (High Sensitivity) Assay Kit (Thermo Fisher Scientific, Waltham, MA, USA). Only samples with a DNA concentration higher than 0.5 ng/μL were included in the analysis. Samples preparation for whole-genome sequencing was conducted in accordance with the Illumina DNA Prep Reference Guide (document #1000000025416 v10) and the LMN DNA LP (M) Tagmentation Kit (Illumina, San Diego, CA, USA). Briefly, 30 μL of each DNA extract was fragmented, and tagged using the transposome included in the kit, with unique indices (DNA/RNA UD Indexes Set A, Tagmentation, San Diego, CA, USA), added to each sample (Appendix A). The tagmentation reaction was incubated at 55 °C for 15 min. Subsequent PCR amplification was performed using the Illumina PCR Master Mix with the following conditions: 68 °C for 3 min; 98 °C for 3 min; [98 °C for 45 s, 62 °C for 30 s, 68 °C for 2 min] for 10 cycles; 65 °C for 1 min. Each library and the final library pool were quantified using a Qubit 4.0 fluorometer. Samples were normalized, pooled, and subjected to paired-end sequencing on an Illumina MiSeq system with a MiSeq V2 cartridge (300 cycles). Sample preparation and sequencing were conducted following the manufacturer’s protocol.

### 2.4. Bioinformatic Analysis

Sequencing data quality was assessed using FastQC 0.12.0, evaluating read counts, maximum/minimum read lengths, GC content, and ambiguous nucleotide proportions for forward and reverse reads.

De novo genome assembly was performed via scaffolding using SPAdes 3.15.5 [21] on the Galaxy server, based on paired-end reads. A taxonomic analysis of scaffolds was conducted using Galaxy’s FCS GX tool 0.5.5 [22] to confirm species identity and remove contaminants. All isolates were confirmed as *A. baumannii*.

Assembled *A. baumannii* sequences were cleaned of residual adapters using NCBI VecScreen: FCS Adaptor 0.5.5 [22] and filtered by scaffold length with Trim.seqs 1.39.5 [23].

Typing was performed via multilocus sequence typing (MLST) and core-genome MLST (cgMLST) using the PubMLST database (https://pubmlst.org/, accessed on 3 June 2025).

Multiple sequence alignment and neighbor-joining (NJ) phylogenetic tree construction were performed in Mauve 2.4.0 [24] using the Progressive Mauve algorithm (default parameters: Full Alignment, Sum-of-pairs LCB scoring, and HOXD matrix). Scaffolds were aligned to the NCBI reference sequence GCF_009035845.1 (https://www.ncbi.nlm.nih.gov/datasets/genome/GCF_009035845.1/, accessed on 10 September 2024).

In silico detection of antimicrobial resistance (AMR) determinants was conducted using ResFinder 4.6.0 [25] (default thresholds: ≥90% identity, ≥60% coverage). Input data consisted of fastq files (paired-end reads).

Genome annotation was performed via the NCBI Prokaryotic Genome Annotation Pipeline 6.8. MLST gene alignment and analysis were conducted in MEGA X 10.2.6 using the MUSCLE algorithm. The whole genome sequences obtained in this study were submitted to GenBank under accession numbers JBHYBP000000000–JBHYCK000000000, JBITPW000000000–JBITRA000000000.

## 3. Results

### 3.1. Isolate Collection and Sequence Types

A total of 56 *A. baumannii* isolates were obtained and included in this study. These isolates represented the entire *A*. *baumannii* population recovered from a large-scale sampling campaign, which included 1,080 environmental and healthcare workers’ PPE swabs and 36 sputum samples. From this collection, a total of 191 bacterial isolates were obtained, with 56 (29.3%) being identified as *A*. *baumannii*. MLST analysis successfully typed 39 isolates, revealing ST2 (*n* = 20, 35.7%), ST78 (*n* = 17, 30.4%), and ST19 (*n* = 2, 3.5%). Seventeen isolates (30.4%) were non-typeable (n/d).

ST2 isolates were obtained from patients (15.0%, *n* = 3) and hospital environmental surfaces (85.0%, *n* = 17), with the majority from PPE (47.0%) and patient care environment (41.2%). ST78 isolates were also predominantly isolated from the environmental samples (82.3%, *n* = 14), with most found on PPE (64.3%). Both ST19 isolates were identified in a patient and on PPE.

### 3.2. Phylogenetic Analysis

Phylogenetic analysis s to the Russian Federation. ST78 isolates showed relatedness to isolates from Greece, Germany, Finland, Denmark, and Belarus. ST19 isolates were similar to isolates from Germany and Spain (Figure 2).

Within the hospital, clear clustering was observed between patient and environmental isolates. For example, a patient isolate (4-II_S19) clustered with isolates from handrails and adjustment levers on an ICU bed, from an orderly’s PPE, and from ICU door handles. Another large cluster included isolates from various surfaces (ventilator, PPE of doctor’s/nurse’s/orderlie’s, and manipulation table), underscoring the role of healthcare workers in transmission.

### 3.3. Antimicrobial Resistance Determinants

Resistance genes to eight antimicrobial classes were identified: aminoglycosides, β-lactams, macrolides, streptogramin B, amphenicols, rifamycins, antifolates, and tetracyclines.

All STs showed 100% prevalence of resistance genes to aminoglycosides and β-lactams.

ST2: Characteristic profile: *armA*, *aph*(*6*)-*Id*, *aph*(*3′*)-*Ia*, *aph*(*3″*)-*Ib*, *bla*OXA-23, *msr(E),* and *mph*(*E*). The *bla*OXA-23 carbapenemase was present in 100% of isolates.ST78: Characteristic profile: *armA, bla*OXA-72, *bla*OXA-90, *msr(E),* and *mph(E)*. *bla*OXA-23 was absent.ST19: Profile included *aph*(*3′*)-VIa, *blaOXA*-*69*, *blaADC*-*25*, *catA1*, *sul2*, and *tet*(*B*).

The n/d group’s resistance profile resembled ST2, suggesting possible genetic relatedness or horizontal gene transfer.

When analyzing genetic profiles, isolates were divided into two groups: 1—patients (Appendix B Table A2); 2—environmental surfaces (Appendix B Table A3). Isolates from both groups represented all detected ST (Figure 3). All ST, including the n/d group, showed 100% presence of resistance determinants to aminoglycosides and β-lactams, with presumed MDR (resistance to 4–5 antimicrobial classes) across all groups.

### 3.4. Virulence Determinants

The in silico analysis revealed a rich repertoire of virulence-associated genes across all studied isolates. The most prevalent functional groups included genes encoding:Efflux pumps:
oATP binding cassette: *macA*, *tolC*, which were nearly ubiquitous (>90% of isolates);oMultidrug and toxic compound extrusion: *abeM* (>90% of isolates);oMajor facilitator superfamily: *abaF* (>90% of isolates);oResistance nodulation division: *adeA*, *adeH*, *adeI*, *adeJ*, *adeK*, *adeN* (>90% of isolates);oProteobacterial antimicrobial compound efflux: *aceI* (>90% of isolates);oSmall multidrug resistance: *abeS* (>90% of isolates);oLipopolysaccharide: *lpxC* (>90% of isolates).
Pili:
oChaperon-usher type I pili: csuAB (>90% of isolates);oType IV pili: *pilT*/*pilU* (>90% of isolates).
Metal ion uptake systems:
oAcinetobactin: *basG*, *basH*, *basJ*, *bauB*, *bauC*, *bauD*, *bauE* (>90% of isolates);omum operon: *mumR* (>90% of isolates);oMetal homeostasis regulators: *fur* (>90% of isolates);oZinc uptake system: *zigA*, *znuB*, *znuC* (>90% of isolates).
Two-component systems:
oAdeRS;oBaeSR;oBfmRS (>90% of isolates);oLPS modification: *pmrB*.
Secretion system:
oT1SS: *hlyD* (>90% of isolates);oT2SS: *gspD*, *gspE*, *gspG*, *gspM* (>90% of isolates);oT4SS: *traC*, *traL*, *traU*, *traV*, *traW*;oT5SS: *ata*;oT6SS: *tssB*, *tssC*, *tssD*, *tssK* (>90% of isolates).
Miscellaneous:
oImmune evasion: *tuf* (>90% of isolates);obiofilm development: *recA*;oIn vivo survival, killing of host cells: *gigA*, *gigB*, *gigC*, *gigD* (>90% of isolates);oCsu Pili expression: *cheA*, *cheY*;oNeutrophil recruitment: *paaE* (>90% of isolates);oKilling of host cells: *ompR* (>90% of isolates);oSerum resistance, invasion: *cipA*;oSerum resistance, in vivo survival: *surA1* (>90% of isolates).


A complete overview of the detected virulence genes, grouped by functional category and sequence type, is provided in Appendix A.

### 3.5. Microevolution and SNP Analysis

An analysis of classical MLST genes for ST2 revealed complete identity with reference sequences. However, cgMLST analysis assigned most isolates to a common cgST-1746 but revealed significant heterogeneity, with allele mismatches ranging from 24 to 583 (Table 1). Isolate 188-II_S28 had the highest number of mismatches (583) and contained a unique synonymous SNP (T2220G) in the *rpoB* gene, not previously reported in databases.

## 4. Discussion

One of the most prevalent nosocomial pathogens in modern healthcare settings is *A. baumannii*, which can cause urinary tract infections, bacteremia, meningitis, pneumonia, and catheter-associated infections [26,27]. Immunocompromised systems, particularly those with burn injuries or those admitted to the ICU, are at the highest risk of infection. The recent COVID-19 pandemic was associated with an increased incidence of ventilator-associated nosocomial secondary infections, for which *A. baumannii* served as a primary etiological agent [28,29].

According to the 2019 Global Antimicrobial Resistance and Use Surveillance System (GLASS) report, 132,000 *A. baumannii* infections directly contributing to patient mortality were resistant to at least one clinically important antibiotic, with 57,700 fatal cases demonstrating carbapenem resistance [30].

The global prevalence of CRAB continues to rise relentlessly, severely limiting treatment options and exacerbating morbidity and mortality rates associated with these infections.

Key mechanisms of carbapenem resistance in *A. baumannii* include enzymatic inactivation, porin modifications, efflux pump upregulation, and β-lactamase production. Among the four known β-lactamase classes (A–D), classes B and D are primarily responsible for carbapenem resistance.

OXA-type carbapenemases (Class D) represent the most prevalent carbapenemases in *A. baumannii*, particularly blaOXA-23-like enzymes. These enzymes are typically plasmid-encoded, facilitating horizontal gene transfer and dissemination within bacterial populations, especially in the hospital environment. The blaOXA-23-like carbapenemase was the first OXA-type carbapenemase identified in CRAB and remains the most globally widespread variant [31]. Consequently, it represents a critical target for molecular surveillance, whose integration into HAIs epidemiological monitoring systems is imperative.

In this study, we showed that ST2 is the predominant lineage in the ICU, with isolates from 3 patients and 17 environmental surfaces (8 from PPE, 7 from patient care areas, and 2 from general hospital sampling points). The prevalence of ST2 reflects the evolutionary success of this clonal lineage (CC2), which accounts for most sequenced genomes globally. The characteristic multidrug resistance profile of ST2 lineage, coupled with limited therapeutic options, ensures its continued prevalence in healthcare facilities [32].

A cgMLST analysis revealed a single prevalence cgST-1746 for most isolates. Isolate 104-II_S26 exhibited both cgST-1746 and cgST-11006, while isolate 188-II_S28 (max mismatches = 583) contained a unique *rpoB* SNP, potentially indicating accelerated evolutionary adaptation. The mismatch distribution showed no temporal correlation with sampling chronology. The observed cgMLST heterogeneity despite conserved cgST is characteristic of long-term hospital-adapted populations and reflects the remarkable genomic plasticity of *A. baumannii*.

ST78 represented the second most frequently detected ST, with 3 patient isolates and 14 environmental isolates (9 from PPE, 3 from patient care environment, and 2 from hospital sampling points).

ST19 isolates were sporadically detected and did not significantly impact the epidemiological landscape of the COVID-19 ICU.

Beyond antimicrobial resistance mechanisms, *A. baumannii* pathogenicity is mediated by diverse virulence factors. While all major STs shared this “core” set of virulence genes, future studies on a larger sample size could elucidate potential subtle differences in virulence profiles between sequence types and their correlation with resistance patterns. Nevertheless, the convergence of MDR (e.g., *blaOXA*-*23*) with this robust arsenal of virulence factors underscores the threat posed by these strains and their ability to cause difficult-to-treat infections in a critical care setting.

Phylogenetic analysis incorporating isolates from the PubMLST international database revealed clustering patterns associated with specific geographic regions, suggesting potential pathogen introductions.

Within the infectious disease hospital, distinct clustering was observed between isolates from patients, medical workers’ PPE, and patient environment surfaces. For the two predominant lineages (ST2 and ST78), PPE represented the most contaminated sites (47.0% and 64.3% of isolates, respectively), underscoring the critical importance of strict infection control measures and hygiene practices in healthcare facilities to prevent the spread of antibiotic-resistant bacteria.

In silico analysis identified resistance determinants to eight antimicrobial classes in all *A. baumannii* isolates. Each sequence type exhibited a characteristic resistance profile. All ST2 samples are characterized by a genetic profile of resistance that includes *armA*, *aph*(*6*)-*Id*, *aph*(*3′*)-*Ia*, *aph*(*3*″)-*Ib*, *blaOXA*-*23*, and *msrI* genes. Meanwhile, all ST78 samples have a genetic profile of resistance that includes the same resistance genes as ST2, as well as *blaOXA*-*72* and *blaOXA*-*90*. Separate resistance genes for various AMR classes have been found in ST19 isolates.

Notably, all ST2 isolates carried the *blaOXA*-*23* carbapenemase, confirming their CRAB status, while this determinant was absent in ST78 and ST19 lineages. The results are very similar to results published during the same COVID-19 period in Croatia, where *A. baumannii* isolates carrying *blaOXA*-*23* carbapenemase were identified in ICU patients and air conditioners [33]. This finding is consistent with reports from another Croatian hospital, where CRAB was isolated from hospital surfaces [34], highlighting the role of the contaminated hospital environment in the persistence and spread of these pathogens. Similar results of an *A*. *baumannii* producing *blaOXA*-*23* carbapenemase outbreak in Basel, Switzerland, were reported in an ICU [35].

One study reported an increased incidence of ventilator-associated pneumonia caused by *A*. *baumannii* during the pandemic in Mexico. An increase in resistance to aminoglycosides, carbapenems, folate inhibitors, and other antibiotics was also observed [36]. Furthermore, the detection of genetic determinants of resistance to these antibiotic classes in our samples is consistent with the observed phenotype of multidrug resistance in *A*. *baumannii* strains described during the pandemic period.

WGS and cgMLST emerged as the most informative bioinformatics tools, offering superior resolution for HAIs epidemiological surveillance compared to classical MLST approaches.

This study has several limitations. It is restricted to a single ICU, and our conclusions are based solely on genotypic predictions of antimicrobial resistance and virulence, lacking phenotypic validation through standard microbiological methods. Furthermore, while significant genetic heterogeneity was observed, the study design did not allow for the investigation of specific drivers of microevolution, such as detailed patient antibiotic exposure or environmental factors. Despite these limitations, the obtained results emphasize the significance of high-throughput sequencing for studying pathogen biology. By determining the degree of similarity among isolated strains, identifying their resistance and virulence determinants, and investigating the epidemiological connections between pathogens, we can more accurately identify the key factors contributing to nosocomial infections. These findings will be valuable for developing more efficient strategies to prevent and control infections caused by *A. baumannii* in hospital settings.

## 5. Conclusions

In conclusion, the ICU environment during the COVID-19 pandemic was dominated by MDR *A. baumannii* ST2, harboring the *armA* 16S rRNA methylstransferase and the *bla*OXA-23 carbapenemase. The study demonstrates the critical role of contaminated environmental surfaces, especially PPE, in the transmission chain. The high resolution of WGS and cgMLST is essential for effective epidemiological surveillance and outbreak investigation. These findings emphasize the need for reinforced infection prevention and control measures in healthcare settings managing vulnerable patient populations.

## Figures and Tables

**Figure 1 pathogens-14-00961-f001:**
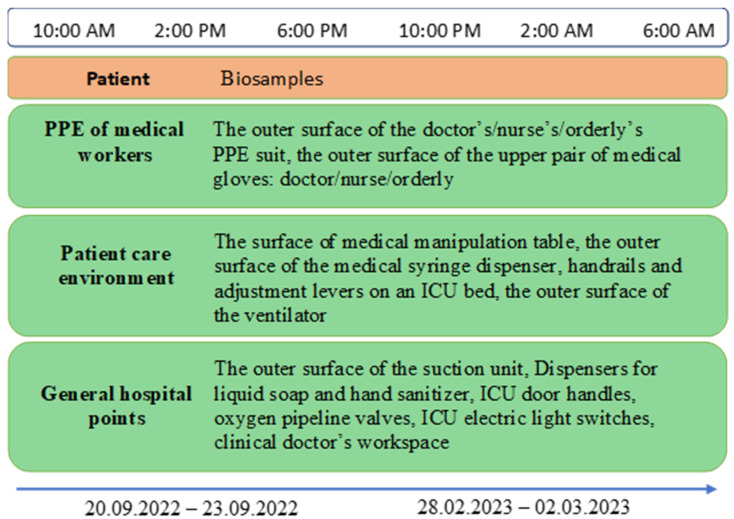
Study design and sampling scheme from patients and hospital environment.

**Figure 2 pathogens-14-00961-f002:**
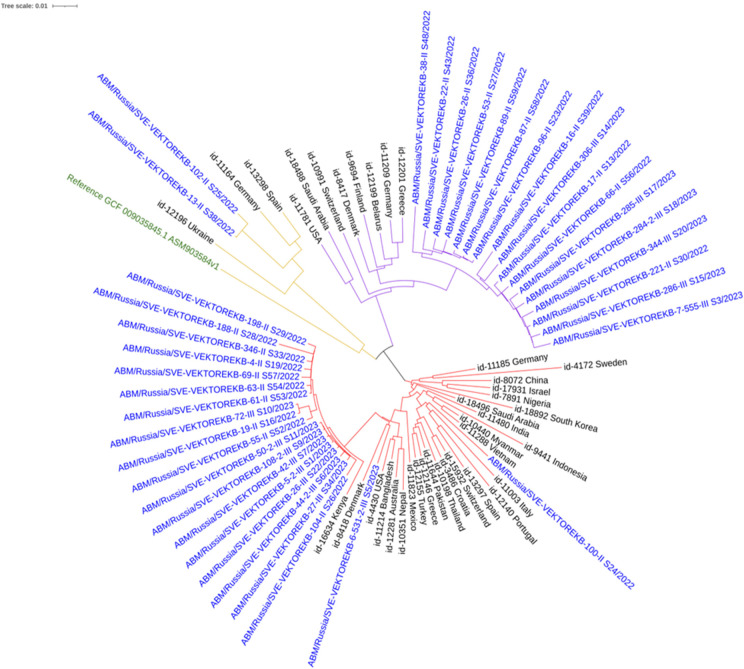
Neighbor-joining phylogenetic tree of *A. baumannii* clinical isolates and reference sequences from the PubMLST database. Colored clades: red (ST2), purple (ST78), and yellow (ST19). Blue: study isolates; Black: PubMLST database; Green: GenBank reference.

**Figure 3 pathogens-14-00961-f003:**
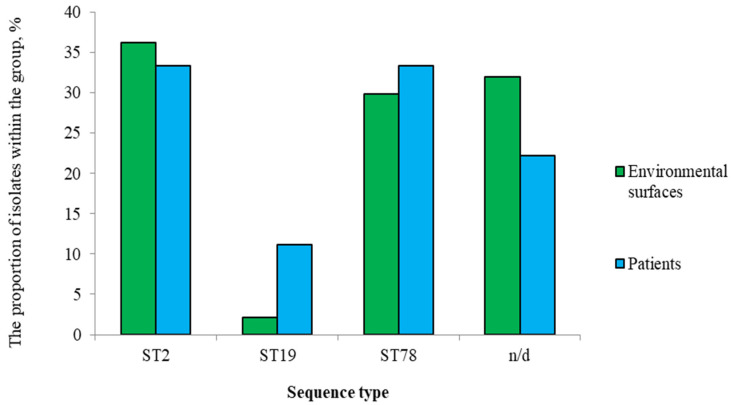
Distribution of *A. baumannii* ST isolated from patients and environmental samples.

**Table 1 pathogens-14-00961-t001:** Comparative analysis of *A. baumannii* ST2 isolates according to the cgMLST scheme.

Sample ID	Closest cgST	Mismatches, *n*	Loci Matched, %
6-531-2-III_S5	1746	24	98.9
108-2-III_S9	1746	110	94.8
50-2-III_S11	1746	115	95.6
61-II_S53	1746	115	94.6
72-III_S10	1746	117	94.5
42-III_S7	1746	121	94.3
26-III_S22	1746	129	94.0
55-II_S52	1746	138	93.5
19-II_S16	1746	156	92.7
27-III_S34	1746	171	92.0
346-II_S33	1746	186	91.3
63-II_S54	1746	202	90.5
44-2-III_S6	1746	211	90.1
4-II_S19	1746	222	89.6
69-II_S57	1746	228	89.3
198-II_S29	1746	232	89.1
100-II_S24	1746	258	87.9
104-II_S26	1746, 11,006	461	78.4
188-II_S28	1746	583	72.7

## Data Availability

The raw sequencing data generated in this study have been deposited in the NCBI SRA database under BioProject accession number PRJNA1165946.

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
