# Peer review of "Genetic Characteristics of Acinetobacter baumannii Isolates Circulating in an Intensive Care Unit of an Infectious Diseases Hospital During the COVID-19 Pandemic"

_pathogens, 2025, doi:10.3390/pathogens14100961_

Round 1
Reviewer 1 Report
Comments and Suggestions for Authors
This paper presents a genetic analysis of Acinetobacter baumannii strains from 2022 and 2023. The authors have thoroughly presented the results, particularly the detailed data in the Appendix.
Unfortunately, the methodology is lacking, including:
1. Detailed data regarding the A. baumannii strains, their origins, and hospitals are missing. If these strains originated outside of Russia, how were they obtained?
2. The authors did not specify the primers used.
3. Information is missing as to whether the strains were cultured prior to DNA isolation, and if so, on what media, under what conditions, and how they were identified.
4. Chapter 2.2 lacks descriptions of the reaction conditions.
5. It is regrettable that only 56 strains were tested.
Reviewer 2 Report
Comments and Suggestions for Authors This manuscript presents a comprehensive investigation into the genetic characteristics of Acinetobacter baumannii isolates from the ICU of an infectious diseases hospital repurposed for COVID-19 patients, with a focus on antimicrobial resistance (AMR) and transmission dynamics. Methodologically, the study demonstrates strong scientific rigor: the integration of whole-genome sequencing (WGS) and core-genome multi-locus sequence typing (cgMLST)—as opposed to only classical MLST—provides high-resolution insights into strain relatedness and microevolution, addressing the well-documented limitation of traditional typing methods in tracking intra-hospital bacterial evolution. The sampling design, covering both patient and environmental sources (including personal protective equipment [PPE], patient care surfaces, and general hospital areas) using a patented swab protocol, is also commendable for capturing the full scope of A. baumannii circulation in the ICU ecosystem.The identification of three sequence types (ST2, ST78, ST19) and the detailed profiling of resistance genes across eight antimicrobial classes further enhance the study’s scientific value. The consistent detection of aminoglycoside and β-lactam resistance genes in all isolates aligns with clinical observations of multidrug-resistant (MDR) A. baumannii prevalence in post-COVID-19 ICUs, and the discovery of a novel synonymous SNP (T2220G) in the rpoB gene adds a unique contribution to understanding strain adaptation. Phylogenetic analyses linking local isolates to global lineages (e.g., ST2 to East Africa, ST78 to Europe) also provide meaningful context for the epidemiological origin of these pathogens.
However, the study has notable gaps in experimental completeness. First, the analysis is restricted to a single ICU within one hospital. While the data are robust for this setting, the lack of multi-center or regional sampling limits the generalizability of conclusions about A. baumannii circulation during the COVID-19 pandemic. Second, despite highlighting the role of PPE in transmission (47.0% of ST2 and 64.3% of ST78 isolates from PPE), the manuscript does not include quantitative data on PPE usage frequency, disinfection protocols, or healthcare worker compliance—factors critical to validating PPE as a "key transmission route" rather than just a contaminated surface.
Another limitation lies in the characterization of virulence factors. The discussion notes that virulence determinants are "incompletely characterized" but provides no preliminary data (e.g., in silico prediction of biofilm-related genes or secretion systems) or plans for follow-up studies. Given that virulence is tightly linked to A. baumannii pathogenicity in immunocompromised COVID-19 patients, omitting this dimension weakens the study’s relevance to clinical outcomes. Additionally, while cgMLST reveals significant heterogeneity (24–583 allele differences) among ST2 isolates, the manuscript does not explore potential drivers of this microevolution—such as selective pressure from antibiotic use (e.g., carbapenem exposure) or environmental conditions (e.g., humidity, surface material)—which would strengthen mechanistic insights into strain persistence.
The in silico detection of AMR genes using ResFinder (≥90% identity, 60% coverage) is methodologically sound, but the study would benefit from correlating genetic resistance profiles with phenotypic susceptibility testing. While the manuscript infers MDR based on genotypic data, phenotypic validation (e.g., broth microdilution) is standard practice to confirm the functional relevance of detected resistance genes, especially for clinically critical determinants like blaOXA-23.
Despite these gaps, the study’s core findings—particularly the utility of WGS/cgMLST for epidemiological surveillance and the role of environmental contamination in A. baumannii spread—are valuable for guiding infection control in post-pandemic ICUs. Addressing the limitations outlined above would significantly elevate the manuscript’s impact.
To contextualize the study within the broader landscape of COVID-19-related secondary infections, we recommend citing two relevant works: 1) "Effects of traditional Chinese medicine on treatment outcomes in severe COVID-19 patients: a single-centre study" (Chin J Nat Med, 2024, 22(1): 89-96), which explores adjunctive therapies for severe COVID-19—populations at high risk of A. baumannii co-infection—and complements the manuscript’s focus on ICU patient vulnerability. 2) "COVID‐19 and cancer: Dichotomy of the menacing dilemma" (MedComm – Oncology, 2023;2:e58), which discusses immunocompromise in COVID-19 patients with comorbidities (e.g., cancer)—a group likely overrepresented in ICUs and susceptible to MDR bacterial infections—thus enhancing the clinical relevance of the current study’s findings on A. baumannii transmission and AMR.
Round 2
Reviewer 1 Report
Comments and Suggestions for Authors
The authors significantly corrected the manuscript according to the reviewer's suggestions. Recently, I recommend the article for publication.